# Determinants of Consumers’ Online/Offline Shopping Behaviours during the COVID-19 Pandemic

**DOI:** 10.3390/ijerph18041593

**Published:** 2021-02-08

**Authors:** JiHyo Moon, Yunseon Choe, HakJun Song

**Affiliations:** 1Institutional Research Team, Office of Graduate School, Korea University, 145, Anam-Ro, Seongbuk-Gu, Seoul 02841, Korea; tourism88@naver.com; 2School of Community Resources and Development, Arizona State University, 411 N. Central Ave, Phoenix, AZ 85004, USA; Yunseon.Choe@asu.edu; 3The Hainan University—Arizona State University Joint International Tourism College, Hainan University, 58 Renmin Road, Haikou 570004, China; 4Department of Hotel and Convention Management, Pai Chai University, 155-40 Baejae-Ro, Doma-Dong, Seo-Gu, Daejeon 35345, Korea

**Keywords:** COVID-19, online shopping channel, offline shopping channel, ordered logit model

## Abstract

The COVID-19 pandemic has wreaked havoc in Korean society since the end of 2019. Unlike prior to the pandemic, when online and offline activities were conducted side-by-side, many aspects of consumers’ daily lives are only conducted online, especially shopping and meetings. This study analysed the characteristics of consumers who have used offline shopping channels during the pandemic. In addition, participants were asked how often they will use online and offline shopping channels after society stabilizes from COVID-19 in order to analyse what determinants will be used to select either online or offline shopping channels after the pandemic. This study will contribute to provide a deeper understanding of the consumption patterns of consumers (online vs. offline) during times of deep external impact, such as a pandemic.

## 1. Introduction

In the age of the fourth industrial revolution, the buying patterns of consumers have switched from traditional digital purchases to online or mobile channels due to consumers’ easy access to digital technology as well as the availability of world markets with this technology [1]. Smart digital devices and technology have enabled the service industry to provide services with precision and allow consumers to interact with service providers without ever having to meet face-to-face with an employee. These types of non-contact services have recently become a focus for the consumption patterns of consumers due to the unprecedented COVID-19 pandemic. Even before the beginning of the COVID-19 pandemic, untact marketing was a trend in the distribution industry. Untact is the combination of contact and the negative prefix un, thereby creating the word (un + contact), which means not having any physical interaction. Untact can be defined as a form of service exchange that prevents direct contact with providers and consumers, such as restaurant kiosks, VR (Virtual Reality) shopping, chatbots, and other apps with high technology aspects. The most representative forms of untact services are the kiosks at McDonalds, KFC, and multiplexes. Furthermore, mobile food delivery apps, such as Baedal Minjok and Yogiyo, have become increasingly popular in South Korea (hereafter Korea) which is the subject of the current study. The popularity of untact marketing can be attributed to the following factors: an increase in the number of single person households; changes in the population and demographics of consumers; changes in the social climate; and other external factors, such as the COVID-19 pandemic. Recently, the COVID-19 pandemic has increased consumers’ desire to not meet with others and, as such, increases in untact consumerism have been observed, not only in younger demographics, but also in older demographics, where the use of smart phones and displays has become increasingly prevalent. Furthermore, through the transition into the fourth industrial revolution, development in untact technology has gained ground. It was only natural that the distribution and service industries would reflect these changing trends [2]. A heightened perception exists that the 21st century is the age of globalization and pandemics. Through the progress of globalization, much development has occurred in industrialization and the economy; however, it has also had negative effects, such as increases in the quickness and coverage of viruses throughout the world. Different from the past, as people exchange goods and services, diseases could also inevitably be included in this exchange. Even before the COVID-19 pandemic, Korea was experiencing a difficult situation due to Middle East Respiratory Syndrome (MERS) [3]. Specifically, on 20 May 2015, a quarantine system in Korea came into jeopardy when a MERS outbreak occurred. MERS was previously only thought to have been prevalent in the Middle East, but its influx into Korea caused psychological and physical torment to Koreans.

Additionally, the 2015 incident hit the distribution and services industries strongly, causing growth in these sectors to freeze. This incident created a national understanding of the severity of pandemics, and manuals and policies were put into place to prevent such incidents from happening again internationally or domestically; however, despite all of these efforts, it seems that future epidemics could not be completely prevented. To this end, during the COVID-19 pandemic, the Korean government has increasingly taken a larger role to operate a quarantine management system. When panic occurred that resulted in a mass buying of masks, which caused mask supplies to become dangerously low, the government stepped in and prevented online sales and the export of crucial medical supplies. Additionally, the government implemented a five-day system at local pharmacies through which consumers could only buy masks on certain days of the week based on their national registration number. Although some people began panic buying and became hoarders, an increasingly large proportion of the population turned to internet shopping and online food delivery services. These changes caused a heavy burden on the postal services and restaurants where the number of deliveries increased tremendously. The COVID-19 pandemic has caused consumer habits to change from contact consumerism to untact consumerism. Although pandemics are not common occurrences, a single occurrence of a pandemic can cause an unspoken amount of damage. Most analyses of and research trends related to diseases and epidemics are based in the medical or legal fields. As such, little research exists from the social science perspective. Therefore, even though the world is faced with severe threats from this pandemic, it is difficult to find research focused on changes in online and offline shopping patterns. As such, this study tries to analyse the distribution and retail sectors, which have become increasingly competitive, and the changes in the industries brought about by the COVID-19 pandemic.

More specifically, this study aims to analyse the determinant factors based on online and offline shopping patterns by grouping consumers into online and offline shoppers. In this study, various factors affecting the selection of online and offline shopping channels will be compared and analysed. Although this study follows a similar flow as used in previous research, such as [4,5,6,7], it has a unique aspect to it in that it mainly focuses on the use of retail and distribution by consumers related to the COVID-19 pandemic. The study is expected to provide meaningful data because it investigates the development of and changes in competition between the offline and online shopping industry in the post-COVID-19 era. Additionally, this study aims to analyse the retail patterns of consumers during the COVID-19 pandemic through the use of an ordered logit model, which is a sub-section of the binary choice model. Most related studies have analysed the characteristics of consumers using a selected shopping channel through a model estimation regarding the binary choices of selecting or not selecting a certain shopping channel. These forms of analyses can clearly compare characteristics between the consumers of specific retail channels [7]. Based on the results of this study, the retail industry could be supplemented with evidential data to determine which marketing strategies should be employed during such a crisis as the COVID-19 pandemic. Furthermore, this study can also contribute to providing a deeper understanding of the consumption patterns of consumers (offline vs. online) during times of deep external impact, such as a pandemic.

### 1.1. Literature Review

#### 1.1.1. Usage Patterns of Retail Channels

Consumer consumption patterns have changed numerous times in history. For example, approximately 100 years ago, consumption occurred in the form of bartering or purchases made from traveling merchants. Since then, purchases have been made via catalogues; retail or boutique stores; and convenience stores, supermarkets, and department stores. The consumption paradigm now seems to be shifting toward online retail through the Internet. In the case of Korea, major retail corporations such as the Shinsegae Group (i.e., E-mart, Shinsegae Department Store) and Lotte Group (i.e., Lotte Mart, Lotte Department Store) have dominated the retail industry in Korea. Additionally, the competition between these two corporations shifted to the competition between the offline and online in Korean retail sectors. For example, Lotte Mart (offline store) and Lotte Mart Mall (online retail) are in competition although they are retail companies in the same brand. In the 2000s, the increased availability of smart phones dramatically increased consumption via mobile apps [8]. For example, in Korea, a mobile app called “Baedal Minjok” came into the delivery industry as the representative of the untact form of online consumption, where its users can order food delivery through the app without ever making contact with any employee or restaurant. This untact form of online consumption shows that a small number of companies are now dominating the entire retail industry [9].

This paradigm shifted again when the COVID-19 pandemic began, causing a sharp increase in demand for online and untact consumption. K-Prevention is a form of social distancing that is recommended by the national government. It requires one to refrain from external activities, such as meeting with another individual or simply going out of one’s home. Consumers are also decreasing the frequency of their visits to large supermarkets or other offline stores due to the fear that there may be COVID-19-positive individuals there; thus, a large proportion of consumers have shifted to online consumption, causing new sales records for online retail [9]. Offline retail sales, on the other hand, have decreased sharply due to fears of infection causing people to stay at home, refraining from external activities, and social distancing. It was reported that the decline in retail sales from COVID-19 is the second largest since the revision of the retail industry sales trend statistics that occurred in June 2016 [10]. However, it is plausible that this trend has not just been isolated to Korea, but is occurring globally. For example, on June 16, 2020, the sales for Walmart’s online store overtook eBay’s sales for the first time [11]. In the e-commerce sector of the United States, Amazon ranked first with a market share of 38%, followed by Walmart at 5.8%, eBay at 4.5%, Apple at 3.5%, and Home Depot at 1.9% [11]. Amazon’s online sales sharply surged due to the increase in their number of users during the COVID-19 pandemic [11]. Increased online sales reveal that the consumption paradigm shift from offline to online is accelerating due to online retail’s advantage of providing retail experiences from the comfort and safety of one’s home. In terms of research on online and offline shopping patterns, several studies in the fields of marketing and retail have focused on the characteristics of consumers who use specific retail channels. Home shopping, which uses catalogues, has been around for several decades and, as such, has been well-studied [4,12,13,14]. These studies utilized gender, age, education, occupation, and income to study the characteristics of consumers who used catalogues for their retail purchases. The characteristics of television home shopping channel consumers, which first became popular in the 1980s, were analysed by James and Cunningham [15] and Freedberg [16]. The factors that they utilized in their analyses were TV consumption and shopping affinity along with other demographical factors, such as gender, age, and education. In the 1990s, much research was conducted on the expansion of the Internet and the popularity of Internet shopping. For example, several studies analysed demographical variables with other traits such as shopping affinity and perception and understanding of shopping because they usually affect online retail consumption [5,17]. Since the 2000s, retail channels have diversified with the changes in traditional markets and supermarkets in the retail industry. Kim et al. [18] analysed consumers’ selections among traditional and other competing markets—Internet shopping malls and superstores—with the factors of customer satisfaction; physical sizes of the markets; parking capacities; and government policies, including laws and regulations, and found that Internet shopping malls were used more often when satisfaction and education levels were higher. Additionally, the results showed that government policies were found not to be influential over consumers’ selections. Kim et al. [6] analysed factors influencing one’s selection of online and offline shopping channels using a logit model and the factors of individual characteristics, product characteristics, and purchasing circumstances. The results showed that major factors influencing the selection of shopping channels were knowledge of the selected channel, purchasing experience, and the value of consumers. Particularly, consumers who valued their time most tended to shop online, while consumers who had available time usually shopped offline.

In the second half of the 2010s, the competition between online and offline retail channels increased. Lee [7] compared the traits of consumers who used Internet retail channels and the traits of consumers who used teleshopping with a nested logit model. The results showed that married females with high incomes and older people usually used teleshopping, while single females along with younger people generally used online shopping. Furthermore, it was found that consumers who preferred Internet shopping accrued information before their purchases and tended to trust others. Lee’s [7] study was specifically based on the notion that the affinity or characteristics of consumers were different between online and offline shoppers. For example, the foundational notion states that online shoppers cannot see the item itself, thus they must rely on information provided by other consumers through their reviews [19]. This process indicates that these types of consumers are more trusting of others and are not confined to traditional consumption patterns. More specifically, online shoppers are exposed to indirect forms of information from the Internet or TV due to the inability to view the item itself. This variable is widely prevalent in all forms of online retail, which is very different from its offline counterpart. Based on the review of the literature, this study hypothesizes that online and offline consumption patterns will be different based on the consumer’s demographic and sociopsychological traits during the COVID-19 pandemic.

#### 1.1.2. Protection Motivation Theory

The protection motivation theory (PMT), a major theory of the current study, was initially developed to explain one’s personal motivation to respond to threats or dangerous actions. According to the PMT, one’s reaction to certain situations can lead to positive changes through one’s protection motivation to overcome that specific situation [20]. Protection motivation is a powerful desire to protect oneself. Thus, when an individual is exposed to a message that threatens the safety or health of him/herself, a change in the individual’s actions occurs to remove this threat. Thus, the individual is able to protect him/herself with the individual’s changed actions [21]. Threat appraisal, a component of the PMT, is related to perceived severity, which dictates that the extent of a threat is based on the situation. Threat appraisal also includes perceived vulnerability, which is one’s perception of the extent to which one is exposed to a threat [20]. Severity is one’s perceived degree of harm that may be caused from the threat. According to the PMT, an individual will form a protection motivation, which allows the individual to have a positive attitude toward a recommended action and, thus, carry out the action. In this study, severity is the degree of the psychological threat related to the COVID-19 pandemic. Vulnerability is the negative expectation of becoming exposed to the COVID-19 virus. Therefore, when a consumer’s recognition of severity and vulnerability is high, a high likelihood exists that the consumer will experience a significant level of personal threat. His/her protection motivations will lower his/her frequency of outside activities, such as using offline retail, and could impact the consumer’s online retail purchases. Another component of the PMT, coping appraisal, includes perceived response efficacy and self-efficacy, allowing an individual to assess the potency of one’s response. Coping appraisal represents the belief of the individual regarding his/her ability to properly respond to threatening situations [22,23]. Coping appraisal consists of response efficacy and self-efficacy. Response efficacy is an individual’s expectancy that the selection of a recommended action can eliminate a threat. Self-efficacy is trust in the ability of oneself to successfully carry out the recommended action. Response efficacy and self-efficacy increase an individual’s potential to effectively carry out protective measures. Therefore, these attributes will increase an individual’s frequency of offline retail purchases as the individual’s protective action effectively decreases the threat (high response efficacy) and increase the expectation of the successful adaptation of the action (high self-efficacy). In summary, based on the PMT, the consumer will not tend to perform actions, such as pursuit of the offline retail channel, to avoid the threat that may expose him/her to the adverse situation if his/her cognitive level of threat appraisal is high. Additionally, if the consumer has a high cognition of response efficacy and self-efficacy, he/she will be more likely to prefer the offline retail channel by carrying out protective measures, such as using hand sanitizers, wearing masks, and avoiding outside activities.

#### 1.1.3. Theory of Planned Behaviour

The theory of planned behaviour (TPB) was introduced by Ajzen [24,25] and has been widely applied to understand various human behaviours. According to the TPB, the leading factors of human behaviours include attitudes to behaviour, subjective norms, and perceived behavioural control. Specifically, the TPB systematically explains factors that are directly related to behavioural intentions during courses leading up to behaviours. This explanation is the reason why this theory is widely acknowledged as providing a useful theoretical framework for explaining human behaviours [26,27]. The attitude toward behaviour in the TPB refers to learned tendencies related to responding either favourably or unfavourably to certain objects. Subjective norm is similar to the social pressures felt by an individual regarding a decision on whether to conduct a certain action. In other words, subjective norm represents the opinion of the reference group, and stands for an individual’s perception of how most people think about a particular action [28]. Perceived behavioural control refers to a person’s subjective evaluation of the level of difficulty of conducting a certain behaviour. By adding the perceived behavioural control to the TPB, it can explain behaviours that are not within the boundaries of a human’s will [29,30,31], and has also shown a high level of accuracy in forecasting behavioural intentions [25,26,27,28,29,30,31,32].

Therefore, when we apply the TPB variables of attitude, subjective norm, and perceived behavioural control, we are able to explain social concerns and an individual’s tendency and capacity related to outside activities due to COVID-19. If a person has a negative attitude toward COVID-19 and a strong perception of the subjective norm, then he/she will prefer using online shopping channels rather than offline channels. In addition, perceived behavioural control will have a positive influence on using offline shopping channels because behavioural intentions are likely to increase when an individual perceives that his/her capability, opportunity, and resources are sufficient and the barriers to his/her behaviours are relatively insignificant. This indicates that if the perceived behavioural control is relatively high, then the person will prefer online shopping channels; otherwise, the person will prefer offline shopping channels.

## 2. Materials and Methods

### 2.1. Research Method

In order to achieve the goal of this research, we carried out parallel documentary and empirical studies. International and domestic articles and statistics were referenced in order to systematize the background of this study. An empirical investigation was conducted by analysing the collected data using an SPSS statistics package and a STATA econometric statistics package. In terms of specific research methodology, frequency and basic statistical analyses were conducted in order to identify the characteristics of the survey respondents. The feasibility and reliability of the metric variables in the collected data were investigated using Cronbach’s alpha and exploratory factor analysis. Furthermore, an ordered logit model was employed using the econometric statistics package STATA.

### 2.2. Measurement

The research variables in this study were derived using the following process. The variables were largely categorized into sections. The first section of the survey questionnaire includes severity (4 categories), vulnerability (4 categories), response efficacy (4 categories), and self-efficacy (3 categories). The degree of effect that respondents had on the intention of the protection response of the consumers during the study period (April–May 2020) was calculated. This study utilized the framework of an expanded PMT. Specifically, based on prior research, this study additionally considered the four major variables of the PMT (i.e., severity, vulnerability, response efficacy, and self-efficacy) and supplementary variables (e.g., cognition of government policy, negative attitude toward COVID-19, degree of knowledge regarding COVID-19, degree of participation in social distancing). Demographic characteristic questions that are included in a traditional demand model, such as marital status, gender, and income, were asked. Through the variables of attitude (3 items), subjective norm (3 items), and perceived behavioural control (3 items), consumers’ intentions of using certain shopping channels during COVID-19 were measured. A Likert scale (1: totally disagree to 5: strongly agree) was used for the above items except for the characteristics on the population statistics.

Based on the literature review, items for the PMT and TPB were deduced. The severity of the PMT can be considered a key variable in this study. The survey items for vulnerability (i.e., degree of damage caused by COVID-19) were taken from Ruan, Kang, and Song [33]. The items of response efficacy were based on Cheng, Wei, Marinova, and Guo’s [34] research. The items about self-efficacy were based on Lee, Song, Bendel, Kim, and Han’s [35] study and they were used to investigate the awareness of respondents’ abilities to control results for individuals’ measures for COVID-19. The survey items related to the TPB were borrowed from Ajzen [25] and Lam and Hsu [36]. The awareness of government policy variable was adapted from Ruan, Kang, and Song [33]. The demographical characteristics were selected based on Lee’s [7] study. The protection behaviours were measured by the frequency of the participant’s visit to online markets and offline markets. We categorized the frequency related to visiting offline and online markets into infrequent (0), neutral (1), and frequent (2) for measurement. Since the consumers were placed into three groups based on their frequency of visiting offline markets, an ordered logit model was employed in our study.

### 2.3. Data Collection

The research data for this study were collected via an online survey conducted during April and May 2020. The respondents were required to be at least 20 years old. In terms of conducting a survey, this study adopted a quota sampling regarding age, gender, and regions based on census data so as to prevent distortions in the survey results due to population characteristics. The overall response rate for this survey was 92.9% (i.e., 251 completed surveys from the 271 customers contacted). Of the responses, 34 questionnaires were eliminated due to incomplete responses or irregularities. As such, 251 questionnaires were coded and used for the analysis.

### 2.4. Research Model

As the dependent variable of this study is ordered, an ordered logit model was used. An ordered logit model usually analyses the response of dependent variables which is ordered, ranked, and hierarchical through a regression equation. As such, the ordered logit model is a more developed form of a traditional regression analysis [37]. Compared with the ordered logit model, the affecting factors in the logit model can only be identified within the two categories of the dependent variable: 0 or 1 when the logit model is used. However, the dependent variables can be identified within three or more different ranked categories in the ordered logit model. Generally, the dependent variable of an ordered logit model does not take hierarchy; it is only defined as an ordered form of data. Hierarchy is defined as something needing a response that is dependent on another response. In terms of a research item, order is needed when the response is not dependent on another response and has an equal position among all the responses; thus, order is defined as the “order” in which the response proceeds from one response to another response for a research item in a questionnaire. The ordered logit model can handle casual effects models for qualitative as well as quantitative independent variables with a discrete dependent variable. In that, traditional regression methods cannot consider discrete responses [38]; when a traditional regression model is applied to an ordered response, it is simply used to calculate the average or conduct a regression on the response number. However, when a designated number is used (i.e., when an ordered Likert-scale is used), the predilection of the respondents cannot be determined if the average is calculated to be 2.5. An ordered logit model can solve this issue by using probability [38]. Therefore, a need exists to utilize an ordered logit model in this study so as to analyse various ordered, dependent variables. Thus, the model that will be used for this study is as follows.
(1)$$y^{*}=x^{\prime }\beta +\varepsilon$$

The $y^{*}$ in Equation (1) is a latent dependent variable (frequency of online and offline retail channel use in this study) and, proposing the standard, the final observed response. $x^{\prime }$ in Function (1) is the determinant (explanatory variables) and ε is the margin of error. If there are numbers present for the value, then the latent and observed responses are as follows.
(2)$$y=\left\{\begin{array}{c} o\;if\;y_{j}^{*}\leq 0\\ 1\;if\;0<y_{j}^{*}\leq \mu_{1}\\ 2\;if\;0<y_{j}^{*}\leq \mu_{2}\\ \cdots \\ L\;if\;\mu_{L-1}<y_{j}^{*}\leq \mu_{L} \end{array}\right.$$

In Function (2), above the μ is defined as the range value of the $y^{*}$. Thus, l is the most applicable value within the range values among L number of observable responses and the former description on the Function (2) is defined as the statistical notion held by the dependent variable in the ordered logit model. Therefore, the probability that y will select the specific value l is as follows.
(3)$$Pr\left(y^{*}\leq \mu_{L}\right)$$

Based on Function (3), the final ordered logit model equation used to analyse the relationship among the variables used in this study is as follows.
(4)$$Y=\alpha +\beta_{1}X_{1}+\;\beta_{2}X_{2}+\beta_{3}X_{3}+\beta_{4}X_{4}+\beta_{5}X_{5}+\beta_{6}X_{6}+\;\beta_{7}X_{7}+\beta_{8}X_{8}+\beta_{9}X_{9}+\beta_{10}X_{10}+\beta_{11}X_{11}+\varepsilon$$

In the above equation, *Y* denotes the following dependent variables respectively: the frequency of using offline shopping channels in the past and the frequencies of intended use of either online or offline shopping channels in the future. *Xn*, respectively, denotes severity, vulnerability, response efficacy, self-efficacy, attitude, subjective norm, perceived behavioural control, level of complying with social distancing, knowledge of COVID-19, and recognition of government policy. Finally, ε denotes the error term.

## 3. Results

### 3.1. Sample Characteristics

Table 1 shows the characteristics of the respondents in this study. Gender was similar with 49.8% male and 50.2% female. The age of the respondents was most prevalent as in their 40s (31.5%), in their 20s and 50s (23.5%), and 21.5% were in their 30s. For marital status, 60.6% of respondents were married. For education, 45.8% were college educated. For occupation, the majority of respondents were office workers, small business owners, or self-employed. Income was evenly distributed. As 60% of the respondents were married, a three-person household was the most prevalent with 39.8%, followed by a two-person household with 33.9%, and a single person household at 26.3%.

In addition, 41.8% of perceptions of COVID-19 were reported to be positive—the COVID-19 situation would improve in the future; 58.2% said that the COVID-19 situation will change more negatively in the future (see Table 2). For the category of the most severe disease or epidemic issue, the COVID-19 issue was the highest with 70.1%. It was observed that the majority of the respondents were participating well in social distancing. A total of 67.7% of respondents were found to not be interested in visiting offline retail channels in the future.

### 3.2. Results of the Ordered Logit Model Analysis

By employing major variables from the PMT and TPB, this study analysed how consumers’ behaviours related to online and offline shopping channels were different from each other during the COVID-19 crisis. Due to social distancing to curb COVID-19, sales revenue for offline markets, such as large grocery markets and department stores, fell significantly, while the sales revenue for online shopping soared. The pandemic has restrained—through social distancing and refrained outside activities—consumers from using offline distribution channels, where contact among consumers and salespersons is constantly occurring. Regardless of these circumstances, it is the purpose of this study to analyse the sociopsychological characteristics of the consumers who usually conduct shopping in offline markets and compare them to customers who usually conduct shopping in online markets. To achieve the purposes of this study, we estimated three dependent variables: frequency of using offline shopping channels during the COVID-19 crisis, intention to use online shopping channels after the stabilization of COVID-19, and intention to use offline shopping channels after the stabilization of COVID-19. Since the frequency of using offline shopping channels during the COVID-19 crisis was measured as “0 (will not nearly use it or have not nearly used it), 1 (will use it when necessary or have used it several times), and 2 (will use it a lot or have used it a lot)”, we used the ordered logit model. The goodness of fit of the ordered logit model can be verified by the likelihood ratio chi-square test, pseudo-R^2^, etc. First, when we analysed the models with the ordered logit model, we found them to be statistically significant at the significance level of 1%. The likelihood ratio chi-square test examined changes in likelihood (or goodness of fit) for each model that included only constants or independent variables. All of the models for this study had goodness of fit since they had statistical significance at the significance level of 1% [38]. Since the t-value of the boundary value, which distinguishes frequency of using a distribution channel, had significance at the significance level of 1%, models 1, 2, and 3 were statistically fit as ordered logit models. A summary of the analysis of the ordered logit model can be found in Table 3.

First, the analysis results of Model 1 for using offline shopping channels during the COVID-19 crisis are as follows (see Table 4). For using offline shopping channels during the COVID-19 crisis, males (*p* < 0.10) in their 20s and 30s (*p* < 0.05) were found to have statistical significance. The major variables of the TPB were not found have statistical significance. Meanwhile, vulnerability (*p* < 0.05) of the PMT had a negative influence (−), and response efficacy and self-efficacy (*p* < 0.001) of the PMT had a positive influence (+). Level of compliance with social distancing was found to have negative influence (−) on using offline shopping channels during the COVID-19 crisis (*p* < 0.001).

Second, the analysis results of Model 2 for using offline shopping channels after the stabilization of COVID-19 are as follows (see Table 5).

For using offline shopping channels after the stabilization of COVID-19, individuals in their 20s and 30s (*p* < 0.05) were found to have statistical significance. Similar to Model 1, the major variables of the TPB were not found to have statistical significance. Both severity and vulnerability (*p* < 0.05) of the PMT were found to have a negative influence (−). Knowledge of COVID-19 (*p* < 0.01) had a positive influence (+) and the recognition of government policy (*p* < 0.05) had a negative influence on using offline shopping channels (−). Finally, the analysis results of Model 3 for using online shopping channels after the stabilization of COVID-19 are as follows (see Table 6). Both attitude and subjective norm (*p* < 0.001) of the TPB had a negative influence (+) on the intention to use online shopping channels. However, perceived behavioural control (*p* < 0.05) was found to have a positive influence (+) on the intention. Both severity and vulnerability (*p* < 0.05) of the PMT were found to have negative influences (−) on the intention to use online shopping channels. Both response efficacy and self-efficacy (*p* < 0.05) were found to have a positive influence (+) on the intention to use online shopping channels. In addition, knowledge of COVID-19 (*p* < 0.01) and recognition of government policy (*p* < 0.05) were found to have a positive influence (+) on the intention to use online shopping channels.

## 4. Discussion and Limitations

### 4.1. Discussion

Due to several factors such as the fourth industrial revolution and other technological developments, untact online sales are increasing. Adding to the trend, the unexpected COVID-19 pandemic has made untact shopping indispensable. As a result, the sales of offline distribution channels are gradually decreasing and the sales of online distribution channels are rapidly increasing, thereby the structure of competition in the retail sector is significantly changing. Based on these circumstances, this study intended to study how online and offline shopping behaviours have changed during the unprecedented COVID-19 pandemic. The government of Korea has implemented various policies to fight COVID-19, such as social distancing, compulsory mask wearing when using mass transportation, and scanning QR codes when visiting public facilities. Additionally, the prosperous trend to significantly reduce direct contacts among people was burgeoning while working at home and video conferencing were becoming the daily norm. In this study, an ordered logit model was used for these analyses since the respondents’ answers ranged from 0 (not likely to use) to 2 (likely to use a lot).

Specifically, in the first model, in terms of using offline distribution channels during the COVID-19 pandemic, it was found that men in their 20s and 30s tended to use offline distribution channels. Moreover, response efficacy and self-efficacy were found to have positive influences on the use of offline distribution channels. However, the vulnerability of the PMT and the level of practicing social distancing negatively influenced using offline shopping channels. These results showed that people who did not comply with social distancing usually used offline shopping channels and vulnerability in the PMT diminishes one’s use of offline shopping channels. This also indicates that the more that people recognized that they are vulnerable under current circumstances [36], the less likely they are to use offline shopping channels. In addition, it seems that the more they thought they could defend against threats like COVID-19, the more likely they were to use offline shopping channels. Based on the results of Model 2, in terms of using offline distribution channels after stabilization of COVID-19, it was analysed that people in their 20s and 30s are expected to use offline shopping channels. Both severity and vulnerability of the PMT and recognition of government policy are found to negatively affect using offline shopping channels. However, knowledge of COVID-19 positively affected using the channels. These results implied that when the circumstances of COVID-19 are severe and people think they are vulnerable, offline shopping decreases. In addition, it was revealed as people’s disregard of government policy increased, they seemed to use more offline shopping channels. Moreover, when people have enough knowledge of COVID-19, they actively perform self-protection measures, which leads to more online shopping. It was interesting that attitude, subjective norm, and perceived behavioural control did not have statistical significance in both Models 1 or 2. This shows that the TPB was not enough to explain consumers’ behaviours during the COVID-19 crisis.

In the third model, with regard to using online shopping channels after the stabilization of COVID-19, it was found that none of the characteristics in the population statistics had any statistical influence. Both the severity and vulnerability of the PMT were found to have a negative influence on using online shopping channels. This result shows that when circumstances are severe and vulnerable as with COVID-19, it is more likely that consumption through offline distribution channels will decrease. The model also showed that response efficacy and self-efficacy were found to have a positive influence on intention to use online shopping channels. This result implies the confidence and feeling that they can cope with crises like COVID-19 increase use of online shopping channels. Knowledge of COVID-19 and recognition of government policy were found to have a positive influence on intention to use online shopping channels. Therefore, we can expect that consumers are more likely to use online shopping channels if they have enough knowledge of COVID-19 and positive recognition of government policy. However, both attitude and subjective norm have a negative influence on the intention to use online shopping channels. These results indicated that when consumers are more negative against circumstances like COVID-19, they are more likely to use online shopping channels. Consumers are also expected to use online shopping channels more when the people around them are negative against using offline shopping channels. During the COVID-19 crisis, while consumers conducted behaviours to protect themselves, they were also concerned about the negative reactions of those people around them regarding their own behaviours, such as not wearing masks. As such, perceived behavioural control was found to have a positive influence on intention to use online shopping channels (i.e., when consumers believe that they can control their behaviours, they are more likely to use online shopping channels). Summarizing the results of this study, we can expect that consumers who do not meticulously practice social distancing, are individuals in their 20s and 30s, or confident that they can effectively cope with COVID-19 are very likely to use offline shopping channels. In addition, we can expect consumers who have knowledge of COVID-19, have positive recognition of government policy, and are confident that they can effectively cope with COVID-19 to use online shopping channels. The major variables that influenced the consumers’ selection of offline distribution channels or online shopping channels were social distancing, recognition of government policy, attitude toward COVID-19, and subjective norm. Consumers who had negative attitudes toward COVID-19 and believed that the people around them might respond to their behaviours negatively indicated that they would use more online instead of offline shopping channels.

From the social psychological perspective, this study examines the characteristics of consumers who use offline or online shopping channels through the PMT and TPB and shows that consumers against the COVID-19 pandemic are more likely to use online shopping channels than offline channels. In this study, the findings revealed that the PMT is primarily more appropriate than the TPB in interpreting consumer behaviour under COVID-19 circumstances. This suggests that the PMT is a theoretical approach that can better explain consumer behaviour than the TPB under serious risk and the unprecedented pandemic situation. It also indicates that, consistent with the previous studies, consumers are trying to take sufficient measures to protect themselves through “protective actions” on the issues of fine dust, environmental pollution, privacy, and health. In terms of comprehensive implications for marketers, based on the results of the current study, it is necessary to comply with social distancing guidelines and encourage consumption activities through online rather than offline shopping channels as COVID-19 is still prevalent in our society. In addition, as consumers who become familiar with government regulations are likely to use online shopping channels than offline shopping channels, government officials should do their best to earn the trust of consumers and raise awareness of consumer protection issues regarding health and safety. In other words, if government policies are well implemented and make consumers trust them more, they will prefer online shopping channels. Therefore, government officials need to continue to seek opportunities to increase consumers’ trust in government policies to effectively cope with COVID-19.

Online and offline shopping channels generally have a complementary relationship rather than competition. In the new era of retail, many consumers have moved from offline retail channels to safe and convenient online channels. Consumers have selectively used offline or online shopping channels depending on the consumers’ specific needs before COVID-19; however, in a situation where COVID-19 is likely to last for a long time, consumers are expected to actively use online shopping channels in a short period of time. As the government has implemented further strengthened social distancing rules, the role and revenues of offline retail channels are further decreasing due to government policy. Since the first case was identified in Wuhan in December 2019, COVID-19 has continued until 2021. Although online channels seek continuous increases in revenues amid the prolonged situation of the COVID-19 pandemic, offline channels need to seek different ways such as closing down stores, reducing spaces, shifting spaces with different usages, and so on. Offline retail channels should also make their survival strategies continuously such as developing entertainment of the space, new space design by combining the fourth industrial revolution technologies and customers’ needs, role shift of purchase to the space of experiences, and so on. In this regard, this study has meaning to reveal that the PMT explained the pandemic situation better than the TPB, which was a well-known theory on consumers’ behaviours under the unprecedented pandemic situation. Like Korean cases, the retail apocalypse also happened in the United States due to the COVID-19 pandemic, shutting down retailers such as J.C. Penny, Sears, Neiman Marcus, and so on. Because of this situation, online shopping has grown in importance compared to offline shopping, but it should not be overlooked that offline retail channels still have value that online channels cannot match. Since offline distribution channels have the aspects of the offline retail experience as a part of the national economy, it is necessary to find a direction to carry out economic activities combining online and offline shopping. In offline distribution channels, consumers can see and touch in real time, have a real-world offline experience, and purchase niche products not available for online purchase (i.e., liquor, gourmet foods, art supplies). The space of the offline retail channel can be moved to the real world such as “showrooms” and “meeting places”. In addition, offline shopping channels need to diversify the functions of store spaces to survive declining demand for offline retail spaces. The government’s support for this can help shift the balance of the retail market between online and offline channels.

### 4.2. Limitations and Future Research

This study has a limitation in that it was conducted during April and May 2020, when COVID-19 was still rampant in society. Since confirmed cases of COVID-19 are still occurring, this time period could be a limitation for this study. Therefore, it is necessary to conduct another study after COVID-19 is gone from society in order to determine whether changes have occurred in consumers’ decision processes. Although this study was conducted by focusing on online and offline shopping channels, it could also be meaningful to attempt to conduct similar studies focusing on more specific subsections of the channels, such as department stores, large grocery stores, and convenience stores. In terms of the development process of the incident, research comparing the characteristics and behaviour of actual consumers before, during, and after COVID-19 needs to be conducted in the future. It will be meaningful for future research to be performed in other countries affected by COVID-19 in order to increase the likelihood of generalization. It will be a very interesting scholarly endeavour to examine the impact of protection motivations and consumer sociopsychological characteristics on shopping behaviour during COVID-19 through examples from different countries. Finally, future investigations need to use social network modelling to more diversely examine consumers’ intentions of using certain shopping channels during COVID-19 and future virus disease outbreaks.

In spite of these limitations of this study and additional future research topics, the current study can be said to be of high academic value in that it analysed the characteristics of consumers who used offline shopping channels during the pandemic. In other words, the findings of this research add to better understanding of the consumers’ perception on the retail channels at the initial COVID-19 pandemic situation by conducting research between April and May 2020 during the expansion phase of COVID-19. The study also contributed to investigating how many of these consumers plan to use offline and online shopping channels after the stabilization of COVID-19 and analysed the determinants for their shopping decisions. The results of this study have significance in regard to the characteristics, as expressed by the PMT and the TPB variables, of consumers who have used either offline or online shopping channels. The results showed that, during crises like COVID-19, the PMT was more adequate than the TPB for interpretation of consumer behaviours.

## Figures and Tables

**Table 1 ijerph-18-01593-t001:** Demographic characteristics of the respondents.

Characteristic	N (%)	Characteristic	N (%)
Gender		Marital status	
Male	125 (49.8)	Single	152 (60.6)
Female	126 (50.2)	Married	99 (39.4)
Age Group		Education	
19–29 years old	59 (23.5)	High school	40 (15.9)
30–39 years old	54 (21.5)	College	66 (26.3)
40–49 years old	79 (31.5)	Undergraduate	80 (31.9)
50 years old and over	59 (23.5)	Postgraduate	65 (25.9)
Monthly household income (KRW) *		Occupation	
Less than 3 million	29 (11.6)	Businessman/Self-employee	60 (23.9)
3–3.99 million	79 (31.5)	Professionals	50 (19.9)
4–4.99 million	74 (29.5)	Office worker	50 (19.9)
Over 5 million	69 (27.5)	Government employee	25 (10.0)
Types of household		Technician	20 (8.0)
2-person household	85 (33.9)	Sales and service employee	15 (6.0)
3 or more person household	100 (39.8)	Student	15 (6.0)
Single person household	66 (26.3)	Homemaker	13 (5.2)
		Others	3 (1.2)

Note: * United States Dollar is equivalent to 1122 Korean Won (KRW).

**Table 2 ijerph-18-01593-t002:** Awareness of COVID-19.

Characteristic	N (%)	Characteristic	N (%)
Awareness of COVID-19		Participation in social distancing	
Becoming better	105 (41.8)	Very active participation	57 (22.7)
Becoming worse	146 (58.2)	Active participation	93 (37.1)
Issues considered the most severe		Normal	59 (23.5)
COVID-19	176 (70.1)	Low level of participation	25 (10.0)
MERS	35 (13.9)	No participation	17 (6.8)
Smog	34 (13.5)	Number 1 priority for social distancing	
Swine Flu	33 (13.1)	Refrain from meeting others	112 (44.6)
Ebola	29 (11.6)	Wearing a mask	70 (27.9)
*Aphthae epizooticae*	28 (11.2)	Working from home	40 (15.9)
SARS	27 (10.8)	Refrain from religious activities	39 (15.5)
Bird Flu	27 (10.8)	Consumption through non-contact means (delivery app, online retail)	33 (13.1)
Severe Fever with Thrombocytopenia Syndrome	26 (10.4)	Keeping 2 m of space from others	32 (12.7)
African swine fever virus	26 (10.4)	Refrain from using public transportation	31 (12.4)
Zika Virus	26 (10.4)	Refrain from participating in cultural activities	31 (12.4)
Plans to visit offline retail channel in the future		Refrain from traveling	31 (12.4)
Have plans	51 (20.3)		
Do not know (neutral)	30 (12.0)		
No plans	170 (67.7)		

**Table 3 ijerph-18-01593-t003:** Summary of analysis of ordered logit model.

	Model 1 (Determinants for Using Offline Shopping Channels during COVID-19 Crisis)		Model 2 (Determinants for Using Offline Shopping Channels after Stabilization of COVID-19)		Model 3 (Determinants for Using Online Shopping Channels after Stabilization of COVID-19)	
**Gender**	+, male	***				
**Age**	+, in their 20s and 30s	**	+, in their 20s and 30s	**		
**Income level**			−	**		
**Attitude**			+	*	−	***
**Subjective norm**					−	***
**Perceived behavioural control**					+	**
**Severity**			−	*	−	**
**Vulnerability**	−	**	−	*	−	**
**Response efficacy**	+	*			+	**
**Self-efficacy**	+	***			+	**
**Level of compliance with social distancing**	−	***			+	**
**Knowledge on COVID-19**			+	*	+	*
**Recognition of government policy**			−	**	+	**

Note: *: *p* < 0.1, **: *p* < 0.05, ***: *p* < 0.01.

**Table 4 ijerph-18-01593-t004:** Result of analysis by ordered logit model of determinants leading to selection of offline shopping channels during COVID-19 crisis.

	Coef.	Std. Err.	z	*p* > z	
**Gender ***	0.971	0.153	6.36	0.000	***
**Age ***	0.743	0.313	2.38	0.017	**
**Income level ***	−0.054	0.349	−0.16	0.877	
**Attitude**	0.244	2.074	0.12	0.906	
**Subjective norm**	−0.500	0.440	−1.14	0.256	
**Perceived behavioural control**	0.081	2.278	0.04	0.972	
**Severity**	0.405	0.314	1.29	0.197	
**Vulnerability**	−1.205	0.519	−2.32	0.020	**
**Response efficacy**	0.666	0.347	1.92	0.055	*
**Self-efficacy**	1.189	0.452	2.63	0.009	***
**Social distancing**	−0.782	0.301	−2.60	0.009	***
**Knowledge on COVID-19**	−1.054	4.208	−0.25	0.802	
**Recognition of government policy**	−0.196	0.143	−1.37	0.171	
**/cut1**	0.063	1.420			
**/cut2**	4.075	1.452			

Note: (*) dummy variable from 0 to 1; Number of obs = 251, Log likelihood = −181.855, LR chi^2^(13) = 74.78; Prob > chi^2^ = 0.000, Pseudo R^2^ = 0.171; *: *p* < 0.1, **: *p* < 0.05, ***: *p* < 0.01.

**Table 5 ijerph-18-01593-t005:** Determinants leading to selection of offline shopping channels after stabilization of COVID-19.

	Coef.	Std. Err.	z	*p* > z	
**Gender ***	−0.437	0.333	−1.31	0.191	
**Age ***	0.735	0.295	2.49	0.013	**
**Income level ***	−0.329	0.129	−2.55	0.011	**
**Attitude**	0.819	0.489	1.68	0.094	*
**Subjective norm**	0.667	0.412	1.62	0.106	
**Perceived behavioural control**	−2.270	2.122	−1.07	0.285	
**Severity**	−0.513	0.299	−1.71	0.086	*
**Vulnerability**	−3.522	1.952	−1.80	0.071	*
**Response efficacy**	−0.239	0.320	−0.75	0.454	
**Self-efficacy**	−0.437	0.415	−1.05	0.293	
**Social distancing**	0.391	0.276	1.42	0.156	
**Knowledge on COVID-19**	7.254	3.944	1.84	0.066	*
**Recognition of government policy**	−0.330	0.137	−2.41	0.016	**
**/cut1**	1.711	1.356			
**/cut2**	5.068	1.400			

Note: (*) dummy variable from 0 to 1; Number of obs = 251, Log likelihood = −181.855, LR chi^2^(13) = 74.78.; Prob > chi^2^ = 0.000, Pseudo R^2^ = 0.171.; *: *p* < 0.1, **: *p* < 0.05.

**Table 6 ijerph-18-01593-t006:** Determinants leading to selection of online shopping channels after stabilization of COVID-19.

	Coef.	Std. Err.	z	*p* > z	
**Gender ***	0.016	0.412	0.04	0.969	
**Age ***	−0.097	0.358	−0.27	0.787	
**Income level ***	−0.144	0.157	−0.92	0.358	
**Attitude**	−8.770	2.457	−3.57	0.000	***
**Subjective norm**	−1.100	0.516	−2.13	0.033	**
**Perceived behavioural control**	6.052	2.665	2.27	0.023	**
**Severity**	−0.782	0.376	−2.08	0.037	**
**Vulnerability**	−1.408	0.598	−2.35	0.019	**
**Response efficacy**	0.894	0.396	2.26	0.024	**
**Self-efficacy**	1.227	0.523	2.35	0.019	**
**Social distancing**	0.871	0.342	2.55	0.011	**
**Knowledge on COVID-19**	9.188	4.862	1.89	0.059	*
**Recognition of government policy**	0.375	0.168	2.23	0.026	**
**/cut1**	3.903	1.687			
**/cut2**	9.579	1.839			

Note: (*) dummy variable from 0 to 1; Number of obs = 251, Log likelihood = −129.444, LR chi^2^(13) = 232.25; Prob > chi^2^ = 0.000, Pseudo R^2^ = 0.4729; *: *p* < 0.1, **: *p* < 0.05, ***: *p* < 0.01.

## Data Availability

The dataset used in this research is available upon request from the corresponding author. The data are not publicly available due to restrictions i.e., privacy and ethical.

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
