# Peer review of "Determinants of Consumers’ Online/Offline Shopping Behaviours during the COVID-19 Pandemic"

_ijerph, 2021, doi:10.3390/ijerph18041593_

Round 1
Reviewer 1 Report
The paper addresses a very recent topic.
Overall the paper looks good however, in the discussion part, I believe there should be a separate section of "Implications to Marketers" since it seems like the focus is too much on Covit itself.
Reviewer 2 Report
This paper describes the results of a very interesting study focused on two months of purchasing habits and behaviors of South Korean households. The main area of improvement is the discussion of results, which needs to be strenghtened in its connection to the Literature Review.
- What is the main contribution of the study to the PMT and TPB Literature? Please expand on this.
- How is this study different from others based on the results?
- What is the future research agenda?
- What exactly can practitioners learn from the study? The authors mention something toward the end, but it needs a more comprehensive reflection, that goes beyond South Korea.
Other comments:
- Lines 128 and 136: a reference is needed in both cases.
- Table 3: there is no need to report the words "negative/positive".
- Line 113: there is no need to report all the names of brands part of those groups. It takes up too much space and add no value to the understanding of the case. Please delete them or just use 1-2 examples.
Good luck!
Reviewer 3 Report
The authors design a study for investigating consumers' characteristics and behaviors about offline shopping channel during COVID pandemia.
The proposed study is interesting but there are some points that the authors should better discuss.
The authors should be better described the novelties of their study with respect to existing ones. In particular, the author should discuss limitation and cons of the examined approaches. Furthermore, the authors should provide more details and discussion about the obtained results. The Discussion section also needs to be improved by analyzing the outcome of evaluation section.
I suggest to further analyze more recent approaches about the examined topics. In particular, I suggest the following papers to investigate multimedia content for influence diffusion and behavioral analysis:
1) Multimedia social network modeling: A proposal. In 2016 IEEE Tenth International Conference on Semantic Computing, 2016; pp.448-453.
2) Multimedia story creation on social networks. Future Generation Computer Systems, 86, 412-420.
Finally, I suggest to perform a linguistic revision.
Round 2
Reviewer 2 Report
Thanks for addressing all the concerns we had. Great job!
Reviewer 3 Report
I think that the authors have addressed all my concerns.